# B Cell and Antibody Responses in Bovine Tuberculosis

**DOI:** 10.3390/antib13040084

**Published:** 2024-10-09

**Authors:** Laura Inés Klepp, Federico Carlos Blanco, María Mercedes Bigi, Cristina Lourdes Vázquez, Elizabeth Andrea García, Julia Sabio y García, Fabiana Bigi

**Affiliations:** 1Instituto de Agrobiotecnología y Biología Molecular (IABIMO) INTA-CONICET, N. Repetto and De los Reseros, Hurlingham 1686, Buenos Aires, Argentina; klepp.laura@inta.gob.ar (L.I.K.); blanco.federico@inta.gob.ar (F.C.B.); vazquez.cristina@inta.gob.ar (C.L.V.); garcia.elizabeth@inta.gob.ar (E.A.G.); 2Instituto de Biotecnología, CICVyA, Instituto Nacional de Tecnología Agropecuaria, N. Repetto and De los Reseros, Hurlingham 1686, Buenos Aires, Argentina; 3INBIOMED, Instituto de Investigaciones Biomédicas, (Universidad de Buenos Aires-Consejo Nacional de Investigaciones Científicas y Técnicas), UBA-CONICET, Paraguay 2155, Buenos Aires C1121ABG, Argentina; mbigi@fmed.uba.ar

**Keywords:** bovine tuberculosis, *Mycobacterium bovis*, B cells, antibodies, cattle

## Abstract

The development of vaccines and effective diagnostic methods for bovine tuberculosis requires an understanding of the immune response against its causative agent, *Mycobacterium bovis*. Although this disease is primarily investigated and diagnosed through the assessment of cell-mediated immunity, the role of B cells and antibodies in bovine tuberculosis has been relatively undervalued and understudied. Current evidence indicates that circulating *M. bovis*-specific antibodies are not effective in controlling the disease. However, local humoral immune responses may contribute to either defence or pathology. Recent studies in animal models and cattle vaccine trials suggest a potential beneficial role of B cells in tuberculosis control. This review discusses the role of B cells and antibodies in bovine tuberculosis and explores antibody-based diagnostics for the disease, including traditional techniques, such as different ELISA, new platforms based on multiple antigens and point-of-care technologies. The high specificity and sensitivity values achieved by numerous antibody-based tests support their use as complementary tests for the diagnosis of bovine tuberculosis, especially for identifying infected animals that may be missed by the official tests.

## 1. Introduction

Bovine tuberculosis (bTB) is an infectious disease primarily caused by *Mycobacterium bovis*, a bacterium that affects cattle and other animals. *M. bovis* and *Mycobacterium tuberculosis*, the main agents of human tuberculosis (hTB), belong to the *M. tuberculosis* complex, a taxonomic group of *Mycobacterium* species that cause tuberculosis in various animals. Although humans are the primary host for *M. tuberculosis*, this bacterium can also infect cattle [1]. Both hTB and bTB are respiratory diseases with highly similar clinical symptoms and immunopathology that predominantly affect the lungs and lymph nodes. In addition, other members of the *Mycobacterium tuberculosis* complex, such as *Mycobacterium* orygis, can also cause bTB [2,3]. bTB significantly affects the economy owing to productivity loss, trade restrictions, and the costs associated with control measures. Understanding the immune response against *M. bovis* is essential for developing effective diagnostics and vaccines.

The initial encounter between *M. bovis* and the host triggers a non-specific innate immune response, in which γδ T cells, monocytes, natural killer cells, and alveolar macrophages play a fundamental role [4]. Recent studies have demonstrated that γδ T cells, monocytes, and alveolar macrophages develop a form of non-specific memory response known as trained innate immunity [5,6]. The innate immune response primes the adaptive response. Dendritic cells are specialised antigen-presenting cells that initiate adaptive immunity by presenting mycobacterial antigens within the context of major histocompatibility complex (MHC) molecules, along with costimulatory signals and cytokines. Dendritic cells migrate from the site of the infection in the alveoli to the draining lymph nodes to initiate antigen-specific T cell responses [7].

Effective immune responses against mycobacterial infection depend on the development of a Th1-biassed cellular-mediated immune response. The production of cytokines such as TNF-α and IL-12 by dendritic cells or macrophages infected with *M. tuberculosis* or *M. bovis* promotes the development of Th1 cells [8]. Conversely, the Th2 response induced by mycobacterial infection antagonises Th1 responses, thus suppressing cellular-mediated immune responses and enhancing humoral responses as the disease progresses. The Th2 response is characterised by the production of cytokines such as IL-4, IL-5, and IL-10 [8,9].

Numerous studies have elucidated the roles of Th1 and Th2 responses in the protective immunity and pathology of *M. bovis* infection [10,11]. However, the key immune components that vaccination must induce to provide protection against bTB have not yet been established.

Some evidence suggests that memory T cells expressing IFN-γ/TNF-α/IL-2 are crucial for protection against this disease [12]. Additionally, cytokines IL-22 and IL-17 may be useful as prominent biomarkers of protection against bTB [13,14,15]. Specific antibody production following Bacillus Calmette–Guérin (BCG) vaccination is seldom detectable in cattle, despite BCG offering partial protection in this species [16]. By contrast, there is some evidence supporting a protective function for BCG-induced antibodies in humans [16]. Increasing evidence suggests that B cells also play a significant role in defending against *M. tuberculosis* infection [17]. However, the mechanisms underlying B cell and antibody-mediated immune responses in *M. bovis* infections remain poorly explored. Therefore, the advances in understanding the role of B cells and antibodies in *M. tuberculosis* infections lack their counterpart in bTB research in cattle or other animals [18]. Most research on bovine antibodies against bTB has focused on developing diagnostic tools or evaluating these proteins as correlates of protection after cattle vaccination.

This review intends to provide a comprehensive overview of the role of B cells and antibodies in the immunopathology and host defence against tuberculosis and the use of antibodies against bTB for diagnostic purposes.

## 2. General Aspects of B Cell and Antibody Responses in Human Tuberculosis

Previous research has shown that some individuals with latent tuberculosis infection presented high titres of antibodies against arabinomannan compared to individuals with active tuberculosis [19]. The higher levels of these antibodies suggest reduced susceptibility to tuberculosis in humans [20,21,22]. In addition, in children, anti-arabinomannan antibodies may be associated with limited tuberculosis dissemination [23]. Other studies, however, have reported lower [24] or similar levels of anti-lipoarabinomannan in latent tuberculosis compared to active tuberculosis individuals [25].

The correlation between the infection outcome and the titre or antigen recognition of sera from human and nonhuman primates revealed that the bacterial burden of active tuberculosis infection was associated with higher antibody titres and reactivity to a much broader repertoire of proteins [26]. These findings underscore the importance of antigen-specificity in the humoral immune response.

Furthermore, a study by Lu et al. [27] has provided more robust evidence about the functional roles of *M. tuberculosis* antibodies in tuberculous infection in humans. According to this research, latent and active individuals showed distinct *M. tuberculosis*-humoral profiles, with some differences contributing to immune activity against *M. tuberculosis*. The differences between cohorts included the ability of antibodies to mediate Fc effector functions (antibody-dependent cellular cytotoxicity, antibody-dependent natural killer cell activation, and antibody-dependent cellular phagocytosis) by THP1 monocytes (a human cell line) [28], as well as significant differences in antibody glycosylation. In addition, a comparison between latent and active antibodies showed that the former specifically enhanced macrophage activation and reduced bacterial intracellular replication in vitro [27]. Protective antibodies against *M. tuberculosis* were also detectable in healthcare workers from a Beijing hospital treating tuberculosis [29].

Thus, antibodies seem relevant to *M. tuberculosis* control, although protection does not solely rely on neutralisation or enhancement of bacterial phagocytosis. Indeed, ex vivo experiments with BCG-vaccinated human blood samples have demonstrated that antibodies play a role in regulating the cellular immune response [30,31].

B cells are lymphocytes responsible for producing antibodies, presenting antigens, and developing into memory B cells after activation by antigen interaction (Figure 1). The role of B cells in controlling an *M. tuberculosis* infection seems to vary throughout the disease, i.e., from acute to chronic stages, in mice [17]. During chronic tuberculosis infections, B cells may exert a proinflammatory action by restricting the expression of the anti-inflammatory cytokine IL-10, a process that contributes to *M. tuberculosis* control but eventually inflicts lung damage [32,33]. In addition, studies in knockout mouse strains showed that, at the initial stages of *M. tuberculosis* infection, B cells provide a source of the proinflammatory cytokine IL-6 [34] (Figure 1). Thus, evidence demonstrates the participation of B cells in regulating the Th1 response mounted by CD4+ and CD8+ T cells.

B cell aggregates resembling B cell germinal centres occurred in granuloma of lungs from humans, nonhuman primates and mice with tuberculosis [35,36,37,38]. In human lungs, B cells aggregate nearby tissue-damaging lesions characterised by necrosis and cavitation [33]. The identification of these B cell structures in tuberculous granuloma supports the interaction of B cells with various immune cells [20].

## 3. The Humoral Immune Response of Cattle upon Vaccination

A study by Buddle et al. [39] in calves has demonstrated that BCG vaccination can lead to low antibody response to *M. bovis* AN5 culture filtrate in the vaccinated groups and that the levels of these antibodies increased in all groups after a challenge or in most groups after the intradermal tuberculin test (TST) (Figure 2). This is one of the first studies reporting this anamnestic response by sensitization. Thereafter, sensitization has been extensively studied and used as an advantage to develop diagnostic techniques.

In the early 2000s, various studies explored different vaccine formulations as candidates in addition to BCG. These formulations included culture filtrate proteins (CFP) combined with IL-2 or cytosine phosphoguanine (CpG) motifs [40] and DNA vaccines encoding MPB70 and MPB83 combined with recombinant proteins. The evaluation of the humoral response in these cases showed high antibody levels compared to BCG [41] (Figure 2). The immune responses in the DNA MPB70/protein group showed a strong humoral component with an IgG1 isotype bias and weaker IL-2 and IFN-γ responses than the BCG-vaccinated group (Figure 2). MPB70 DNA/protein vaccination did not protect animals and these animals, surprisingly, had the highest pathology scores in the trial [41].

Furthermore, the trial using CFP and IL-2 led to the undesirable effect of increasing the extrathoracic spread of the disease [40] (Figure 2). CFP adjuvated with Emulsigen plus a CpG formulation induced protection at levels comparable to BCG [42]. However, when CFP was adjuvanted with Polygen and inoculated together with a CpG formulation, it failed to protect cattle against bTB [42]. The authors concluded that the combination of both Th1 and Th2 immune responses, along with a strong antibody response, might be the reason why the CFP/Polygen + CpG vaccine failed to induce significant protection against an *M. bovis* challenge (Figure 2). Similar vaccination trials using a heterologous prime-boost vaccination approach with HSP65 DNA and protein showed higher levels of IFN-γ that correlated with an increase in serum-specific IgG2 levels and IgG2/IgG1 ratios [43]. However, this research lacked challenge assays and therefore did not yield information on protection.

More recently, Lyaschenko [44] found that skin testing increased MPB83-specific IgG responses in unvaccinated infected and animals, but not in BCG-vaccinated infected animals. This increase correlated positively with disease severity, bacterial loads, and in vitro IFN-γ production induced by ESAT6. Consistent with these results, a BCG vaccination trial in badgers also demonstrated a positive correlation between the extent of antibody responses and disease progression [45]. When comparing live versus formalin-killed BCG in cattle, researchers found that live BCG promoted the development of central memory cells, while killed BCG led to higher levels of IgG1 and IgG2, which is characteristic of vaccinated but unprotected animals [46] (Figure 2).

A revaccination trial of BCG-vaccinated animals also showed that higher antibody levels did not correlate with protection. This trial consisted of revaccinating different groups of cattle two years after the first vaccination by using biopolyester particles (Biobeads) with ESAT6 and Antigen 85A, CFP, or a second dose of live BCG. The protein vaccines induced weak or non-existent antigen-specific IFN-γ responses but significantly increased the humoral response. Unlike revaccinations with live BCG, these formulations failed to protect the animals against challenge [47].

**Figure 2 antibodies-13-00084-f002:**
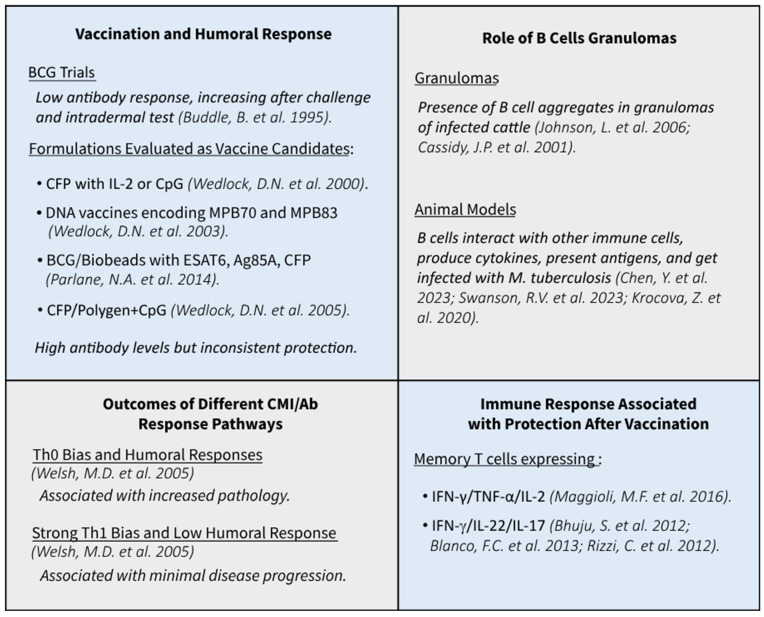
Overview of the immune response against bovine tuberculosis [11,12,13,14,15,33,39,40,41,42,47,48,49,50,51].

## 4. The Humoral Immune Response of Cattle towards *M. bovis* in Experimental Infection

A key study by Welsh et al. [11] analysed the humoral response and the cytokine profile of CD4 T cell clones derived from animals infected with *M. bovis*. The researchers found that a stronger Th0 immune bias and enhanced humoral responses correlate with increased pathology, while strong Th1 immune bias and low anti-*M. bovis* humoral responses were associated with minimal progression of pathology (Figure 2). Increased IgG1 antibody levels and high levels of IL-10 mRNA were also associated with a more widespread pattern of the disease and correlated positively with pathology (Figure 2). In accordance with these findings, IgG1 antibody levels also correlated with pathology and disseminated disease in white-tailed deer experimentally infected with *M. bovis* [52].

Overall, different studies on infection and vaccination in cattle have highlighted certain recurrent aspects of the humoral immune response, i.e., a weak correlation between circulating antibody levels and protective immunity (Figure 2). Local antibody responses likely play more prominent roles in the outcome of *M. bovis* infection, as recently demonstrated in people infected with *M. tuberculosis* [36].

## 5. Role of B Cells in Tuberculosis

The role of B cells in the protective immune response and granuloma formation, aside from antibody production, is an important yet underexplored aspect. Over the past decade, significant findings have emerged on the role of B cells in the immunopathology and orchestration of an effective immune response in the lungs. Although this particular aspect of B cell biology is controversial and less explored in bTB, many studies conducted in mice, macaques and human samples have shed light on this area [18].

Immunohistochemical studies of developing lesions in lung tissue sections from BCG-vaccinated cattle revealed intense clusters of CD79a+ B cells along the outer rim of the granuloma’s fibrous capsule and increased presence of CD79a+ cells within medullary regions, which are distant from the granulomatous inflammation [48] (Figure 1). Similarly, cattle infected with *M. bovis* had large concentrations of B cells in bronchus-associated lymphoid tissues and in the bronchial lymph nodes that appeared later in the infection [50]. These high concentrations of B cells occurred in the periphery of the lesion. In that research, the authors suggested that B cells, together with T cells (WC1+ γδ and CD2+), play a role in the pathogenesis of early-stage lesions [49].

Studies of tuberculosis by *M. tuberculosis* in various animal models, such as mice and macaques, have demonstrated that B cells interact with other immune cells in an antigen-specific manner, produce cytokines, present antigens, and become infected with this bacterium (Figure 1). For example, a study of histological samples from granulomas in nonhuman primate models of aerosol infection found a higher proportion of CD20+ B cells in cynomolgus macaques compared to rhesus macaques, which show a higher disease burden and are considered progressors of the disease [53]. This finding indicates the importance of B cells in controlling tuberculosis. In both species, CD20+ B cells formed follicle-like structures on the periphery of many granulomas in the “organised lesions” categories [53] (Figure 1).

Another study showed that B cell-deficient mice displayed lower levels of lung inflammation—associated with decreased CD4+ T cell proliferation—a diminished Th1 response, and enhanced levels of IL-10. Thus, this evidence demonstrated that B cells modulate protective Th1 immunity and the anti-inflammatory IL-10 response [33].

A recent article published in Nature Immunology describes a deep analysis of B cell function in mice and macaques. In that study, antigen-specific B cells enhanced cytokine production and strategically localised follicular helper T (TFH) cells within granuloma-associated lymphoid tissue via interactions between programmed cell death 1 (PD-1) and its ligand PD-L1, thereby mediating *M. tuberculosis* control [50] (Figure 1).

On the other hand, detailed research on B cells from patients with tuberculosis showed enrichment of CD27 + CD38- memory B cells in lung tissue compared to blood, which was dominated by CD27-CD38- naive and CD27-CD38+ transitional B cells (Figure 1). Additionally, CD27 + CD38hi antibody-secreting cells showed significantly enriched levels in lung tissue compared to matched blood. Antibodies isolated from lung tissue homogenates of patients with tuberculosis confirmed the presence of IgM, IgD, IgG, and IgA reactive to whole *M. tuberculosis* lysate by enzyme-linked immunosorbent assay (ELISA) [36] (Figure 1).

Finally, in vitro cell infection experiments showed that *M. bovis* BCG infects both mouse primary B cells and a human B cell line in a manner dependent on the multiplicity of infection [51]. The intracellular trafficking of BCG inside B cells follows the complete endocytic pathway, which contrasts the events during BCG ingestion by macrophages. Moreover, intraperitoneal murine infection with *M. bovis* BCG activated peritoneal and splenic B cells and produced proinflammatory cytokines [51].

In conclusion, immunohistochemical studies have revealed the presence of B cell aggregates at the periphery of tuberculous lesions in infected and vaccinated cattle (Figure 1). This observation is in line with findings in animal models of tuberculosis. However, whether these B cell aggregates or follicles have a beneficial or detrimental function for cattle is still unclear. In infected animals, the antibody levels increase following the TST, particularly against specific antigens like MPB83. Although this finding indicates the existence of memory B cells, the immune B cell-dependent mechanisms have not yet been studied in bTB.

## 6. Diagnosis of Bovine Tuberculosis

The official antemortem diagnostic test for bTB is the TST. This test involves the intradermal injection of a purified protein derivative from the *M. bovis* AN5 strain. The injection triggers a delayed-type hypersensitivity reaction (type IV hypersensitivity), which results in perifocal inflammation around the injection site. This inflammation, with few or no tubercle bacilli, surrounds a tuberculous focus [54]. Originally developed by Robert Koch in 1890 for human therapy, this procedure fell into disrepute due to its tendency to cause significant tissue damage or reactivate existing tuberculous lesions in some patients [55].

TST can be performed in the mid-cervical region or the caudal skin fold. The Single Cervical Comparative Intradermal Test (SCCIT) compares responses to PPDB and avian PPD (PPDA) to improve specificity [56]. Other diagnostic tests include the interferon gamma release assay (IGRA), which measures the ex vivo production of interferon gamma (IFN-γ) by immune cells following specific stimulation with *M. bovis* antigens, and serological tests that detect the presence of anti-*M. bovis* antibodies in blood samples [56].

Importantly, while the TST evaluates the local immune response at the site of injection, both IGRA and serology tests assess the peripheral immune responses. Therefore, diagnostic methods detecting various aspects of the immune response can be used together to improve both the sensitivity and specificity of the diagnosis.

## 7. Antibodies in the Diagnosis of Bovine Tuberculosis in Cattle and Other Animals

The initial research of Harboe et al. and subsequent studies [11,57,58,59,60] have demonstrated that the cellular response in cattle infected with *M. bovis* is specifically induced a few weeks after infection. By contrast, the production of antibodies to MPB70 or PPDB increases as the cellular response decreases, which typically occurs in the late stages of tuberculosis.

Recently, a study using a very sensitive chemiluminescence immunoassay to study the early dynamics of antibody production of *M. bovis*-infected cattle has demonstrated that a proportion of animals responded with antibody production against a set of mycobacterial antigens as early as three weeks post-infection [61]. Thus, the advent of new immunological technologies and the identification of highly antigenic proteins could challenge the prevailing belief of delayed humoral response against tuberculosis in cattle.

### 7.1. Whole M. bovis Antigen

In 1987, Ritacco et al. [62] developed the first serological test to detect antibodies in *M. bovis*-infected cattle. In their study, an enzyme-linked immunosorbent assay (ELISA) based on PPDB yielded 90% sensitivity and 89.9% specificity when testing serum samples from confirmed *M. bovis*-infected and healthy cattle. Subsequent analysis with a larger number of animals showed lower sensitivity (73.6%) but higher specificity (94.1%) [63] (Appendix A). Another study using whole-cell sonicate as an antigen in an ELISA test showed lower specificity and sensitivity [64]. In all the studies mentioned, negative animals came from bTB-free herds and confirmation of *M. bovis* infection in positive animals was through bacterial cultures. The poorer performance of the test with the whole-cell sonicate could be due to cross-reactions with environmental or nontuberculous mycobacteria or to the absence of *M. bovis*-secreted proteins (which are present in PPDB preparations).

These findings underscore the relevance of *M. bovis*-secreted proteins in the antigen preparations and the need to evaluate serological tests in cattle sensitised to nontuberculous mycobacteria to achieve more reliable specificity and sensitivity values. In fact, Plackett et al. [65] have reported positive reactivity in an ELISA based on culture supernatant proteins of *M. bovis* in cattle exposed to nontuberculous mycobacteria such as *M. avium* or *M. avium subsp. paratuberculosis* (MAP), among others.

### 7.2. Individual Antigens

One year after the first PPDB-ELISA report, a study evaluated an MPB70-based ELISA in different cohorts of cattle infected with *M. bovis* or nontuberculous mycobacteria. The study demonstrated that the production of antibodies against MPB70 is highly specific to *M. bovis* infections [60]. On the other hand, antibodies against the antigen 85 (Ag85) were detectable in cattle infected with other mycobacteria. This is not surprising because MPB70 and MPB83 are not present in the *M. avium* complex, while antigen 85 is highly conserved in the genus *Mycobacterium* and can cause cross-reactions in the analyses [66].

MPB70 exists in non-glycosylated and glycosylated forms, and the glycosylated form is more likely to cross-react with antibodies from nontuberculous animals [67], possibly due to nonspecific reactivity to carbohydrate components of MPB70.

One strategy for developing more precise and sensitive serological diagnostics involves analysing the humoral response after performing the TST. Although this topic has been recently reviewed [18], a study by Waters et al. [68] deserves particular mention. These researchers demonstrated the onset and duration of boosted antibody responses and found that while the skin test enhanced responses to certain antigens such as MPB83 and MPB70, it did not affect responses to other antigens like ESAT6, CFP10, MPB59, and MPB64. The high stability of MPB83 and MPB70 to the heat treatment in PPDB production is likely responsible for this anamnestic response [69,70].

The proteins MPB70 and MPB83 have been the most extensively studied and are now commonly used as antigens for serological tests in bTB because of their immunogenic characteristics [60,71,72,73,74,75,76,77,78,79,80,81,82,83,84,85,86,87,88,89,90,91,92,93,94,95,96,97,98,99,100,101]. Most tests for diagnosing bTB based on antibody detection use these antigens (Appendix A). The sensitivity values of these tests vary depending on factors such as the assay format, the stage of infection, the level of confidence in the animals’ status, the type of infection, and the environmental conditions in the herds (MPB70 and MPB83: <10–100%; Appendix A). In addition, animals infected or sensitised with *M. avium* have shown low or null reactions to MPB70/MPB83, while animals infected with *Mycobacterium kansasii* presented certain levels of antibodies against these proteins [74,85]. These findings indicate reliable performance of MPB70/MPB83 as *M. bovis*-specific antigens.

Ag85A is a member of the A85 complex, which includes Ag85B and Ag85C. These proteins are major secreted proteins of *M. tuberculosis* and *M. bovis* that catalyse transesterification reactions to synthesise mycolated arabinogalactan, trehalose monomycolate (TMM), and trehalose dimycolate (TDM) [102]. Despite Ag85A being part of experimental vaccines against human and animal tuberculosis because of its immunological properties [103,104], its performance as a diagnostic antigen is not promising [44] due, in part, to its high cross-reactivity with nontuberculosis infections [60,66,80]. However, at least two studies have reported acceptable sensitivity (54.2%) and specificity (83.5%) for Ag85A when used in a competitive ELISA [105], and even high sensitivity (91.3%) and specificity (94.8%) in dairy cows from Brazil [106]. The authors suggested that the differences in specificity might be due to the absence of MAP infections in cattle within the studied region [106].

ESAT6 and CFP10 are potent T cell antigens secreted by *M. tuberculosis* and *M. bovis*. These antigens are highly recognized by T cells from infected individuals, making them significant markers for the immune response [107,108,109]. In humans, antibodies against ESAT6/CFP10 have been associated with active tuberculosis [110]. The humoral reactivity of ESAT6 and CFP10 in *M. bovis*-infected cattle has shown high specificity and acceptable sensitivity in diagnostic platforms that combine these antigens with other immunogenic proteins [95,96,99] (Appendix A). When analysed as individual antigens, the dimer ESAT6/CFP10 has shown lower sensitivity than MPB70 and MPB83 but higher specificity [75,93]. Fusion proteins containing peptides from ESAT6 and CFP10 have also shown good sensitivity and specificity when evaluated in TST-positive cattle (sensitivity: 69.4–83.2%, specificity: 86.5–96.0%) [83,84]. Moreover, ESAT6 and CFP10 are valuable antigens since they rarely react with cattle sensitised or infected with nontuberculous mycobacteria [80,111,112,113].

In general, secreted proteins such as MPB70, MPB83, ESAT6 and CFP10 have been preferred as diagnostic antigens because of their high immunogenicity [56,114,115,116]. However, a comparative analysis of immunoreactive *M. bovis* proteins using a panel of 62 serum samples from *M. bovis*-infected cattle has demonstrated, in Western blot assays, that cellular proteins outperform secreted proteins [117]. In addition, by screening an *M. bovis* library in *Escherichia coli* with sera from *M. bovis*-infected cattle, our group identified the membrane-associated protein P27 (LprG) as one of the most reactive antigens [118]. These pieces of evidence support the importance of *M. bovis* cellular proteins in the antibody-mediated immune response regarding bTB. Other studies have confirmed that P27 can induce a powerful humoral response in mice inoculated with a recombinant P27 [119] and in humans with active tuberculosis [110]. Remarkably, P27-immunised mice were highly susceptible to *M. bovis* infection, suggesting that this specific antibody response drives the immune response to an unprotected profile. Therefore, despite the limited information available, the reviewed findings indicate that the cellular proteins of *M. bovis* have potential antigenic value.

The mycobacterial cell wall has various species-specific non-protein compounds, such as glycolipids, long-chain fatty acids, lipoglycans, polysaccharides, and peptidoglycan [120]. However, a few molecules have been evaluated for their effectiveness as diagnostic antigens. Examples of these antigens include LAM, a complex glycolipid that extends from the plasma membrane to the outer surface of the cell wall and can be detected in urine samples [121,122]; phosphatidylinositol mannosides (PIMs), which have shown low specificity [123]; and 2,3-di-O-acyl trehalose (DAT), which has demonstrated low sensitivity [124] (Appendix A).

In conclusion, although the genome of *M. bovis* encodes approximately 4000 proteins, only a few of them have been evaluated as humoral antigens. The MPB70 and MPB83 proteins are among the most immunogenic antigens and with better performance to distinguish between infected and healthy animals effectively. However, the production of antibodies against both proteins in the initial stages of bTB seems to be low [61]. Thus, it is necessary to identify new antigens capable of inducing antibody production in the initial stages of the disease.

In this regard, the Enferplex test (based on MPB70, MBP83, ESAT6 and CFP10) was able to identify few positive animals as early as four weeks post-infection [61]. The early antibodies detected may have been antibodies against CFP10 and/or ESAT6. The proportion of infected animals detected by Enferplex increased after the inoculation of PPDB, possibly due to a boost effect exerted by MPB70, MPB83, ESAT6 and CFP10 present in the PPDB [125].

Tests based on non-protein antigens are in the early stages of development and the results are inconclusive regarding their utility for the serological diagnosis of bTB. In addition, it would be important to consider the cellular proteins of *M. bovis*, which have been overlooked over secreted proteins.

### 7.3. Serological Tests Approved by the World Organisation for Animal Health

Enferplex and IDEXX are serological tests certified by the World Organisation for Animal Health (WOAH, formerly the Office International des Epizooties (OIE)) as suitable for the detection of antibodies to *M. bovis* in cattle serum samples, to be used as ancillary tests. The Enferplex TB method is an assay developed by EnferGroup that incorporates 11 antigens and is currently the only commercially available multiplexed assay to detect bTB in cattle [126]. Other methods based on the same EnferGroup technology with varying numbers of antigens have been reported in the literature, but none of these methods are available commercially [88,96,99,127]. Another serological test commercially available for TB is the IDEXX ELISA, which is an ELISA in a 96-well microtiter plate format that detects antibodies to two *M. tuberculosis* complex antigens (MPB70 and MPB83) [85].

Two independent trials [97,98] have assessed whether combining the single intradermal comparative cervical tuberculin test (SICCT) with the Enferplex TB assay could enhance the detection rates of infected animals. These studies have determined that Enferplex efficiently detects *M. bovis*-infected cattle that were missed by TST. Another study reported similar results employing the IDEXX test [87]. Furthermore, a work published by Casal et al. [82] also concluded that interpreting cellular and antibody detection techniques in parallel is the best approach to obtaining the highest sensitivity and detecting infected animals. The results of these studies suggest that the IDEXX and Enferplex may be adequate as supplementary assays for detecting *M. bovis* infection in areas without routine testing and slaughter.

In naturally infected cattle, both IDEXX and Enferplex have shown improved sensitivity when performed after boosting with PPDB inoculation [99]. Several studies support this boosting effect in different *M. bovis*-infected animals, e.g., in naturally infected goats [128] and experimentally infected cattle [81,100,129,130,131], among others [60,85,132] (Appendix A). Casal et al. [82] performed a study to compare IDEXX with the official diagnostic tests and other serological tests (DR-ELISA, p22_IE and p22_CE). In this case, the researchers did not apply a booster of PPDB before sampling. The IDEXX assay showed low sensitivity to diagnose tuberculosis, similar to a more recent study performed in Northern Ireland [88]. To classify the “true” infection status, the authors used the criterion of infection adopted by Whelan et al. [96]: an infected animal is defined as an animal with a positive result in the skin test, a visible lesion at slaughter and a bacteriological confirmation result. This definition of a positive animal, confirmed by different diagnostic methods, is robust enough to fully trust the reported sensitivity of the serological tests (IDEXX and Enferplex) for diagnosing infected animals. Despite the low sensitivity reported, the authors informed that the tests succeeded in achieving remarkably high specificity.

Another study conducted in Great Britain demonstrated the high accuracy of Enferplex when samples were collected after inoculating PPDB. Using a panel of negative and positive animals (naturally and experimentally infected, as confirmed by necropsy), the researchers found that Enferplex was significantly more sensitive (above 97.9%) than the Bovigam IFN-γ test [96] and that Enferplex did not react with the sera collected from animals vaccinated with BCG. Therefore, Enferplex is suitable as a DIVA test alongside BCG-based vaccine strategies [96].

In Belgium, researchers compared the performance of IDEXX and Enferplex with samples collected after inoculation of PPDB [86]. The multiplexed Enferplex antibody test showed a higher sensitivity for SICCT-positive animals than the ELISA IDEXX test. The interpretation of both serological tests together yielded higher diagnostic specificity, which is crucial for a screening test when the prevalence of bTB is low. Research evaluating both tests using plasma samples taken within the amnestic window, instead of sera, found that Enferplex had higher sensitivity than IDEXX (for SICCT and IFN-ɣ-positive animals) [126]. Although the overall specificity values with the Enferplex test were lower when using plasma samples, the use of this type of sample did not affect the performance of the tests, making it a practicable option for field work.

Taken together, this body of work demonstrates that the Enferplex and IDEXX serological kits can be used as ancillary methods to cell-mediated diagnostic tests in chronically infected herds. Additionally, both tests showed higher sensitivity after the application of PPDB, with Enferplex apparently having better performance.

### 7.4. Serology for M. bovis-Infected Cattle That Escape Cellular Immune Tests

Serological tests for bTB offer the benefit of having the ability to detect infected animals missed by techniques based on cellular-mediated immune responses, such as TST and IGRA. The occurrence of TST-negative animals with lesions containing *M. bovis* confirmed by cultures highlights the need for additional tests [60,65].

Plackett et al. [65] pioneered the development of an ELISA using unheated *M. bovis* culture filtrates to identify anergic-infected animals that did not react to the TST. Subsequently, another ELISA, utilising cellular proteins from the AN5 strain, successfully detected *M. bovis*-infected cattle that tested negative for TST [133]. Further advancements included an ELISA implementation that identified 200 animals as TST negatives, of which 33 were found to have *M. bovis* isolation [134]. Additionally, the dual-path platform (DPP) has proven to be effective in detecting tuberculosis in cattle that did not react to TST [95].

As mentioned above, in the advanced stages of bTB, the T cell response diminishes while antibody production increases. Therefore, using antibody detection as an ancillary test for bTB can enhance the identification of *M. bovis*-infected cattle. It is important to consider, however, that in the reported studies, a low proportion of animals that tested positive for ELISA but negative for TST did not show lesions or *M. bovis* isolation during necropsy. This indicates that the tests may produce false positive results.

### 7.5. Serological Platforms

As mentioned above, ELISA is predominantly used for the serological diagnosis of bTB by using different antigens such as MPB70, MPB83, ESAT6, CFP10, PPDB, and total proteins from cellular and culture supernatants of *M. bovis*. The sensitivity and specificity of ELISA vary depending on the antigens used, the immunoglobulins detected, and the type of biological sample (e.g., blood, urine, milk) (Figure 3). Generally, the specificity is over 90%, while the sensitivity ranges from less than 20% to 100% (Appendix A). However, a few examples in the literature have reported ELISA with positive results for bTB in cattle infected with nontuberculous mycobacteria or sensitised with environmental *Mycobacterium* sp., which may lead to an overestimation of diagnostic performance.

Waters et al. [85] assessed the performance of a commercial test based on the proteins MPB70 and MPB83 in healthy and *M. bovis*-infected cattle (confirmed by PCR or bacteriology), as well as in animals sensitised or infected with *M. avium*, MAP, *M. kansasii*. The results showed a specificity of 98% and a sensitivity of 63%. Using PPDB as an antigen, Ritacco et al. [63] reported an ELISA with a sensitivity of 73.6% and a specificity of 94.1% in bacteriologically confirmed *M. bovis*-infected cattle and in herds settled in a bTB-free country. Importantly, the study included animals infected with MAP, which were mostly negative for the test.

Luminex technology has been applied to diagnose bTB [75,93] (Figure 3). This test allows the simultaneous detection of antibodies against multiple antigens. The antigens are covalently linked to colour-coded fluorescent bead sets, which are combined in the same wells, and sera are incubated in the wells to identify antigen–antibody interactions by using a dual laser system based on the principles of flow cytometry. Two studies with Luminex for tuberculosis in cattle reported high specificity (>95%) and sensitivity over 40% when three positive antigens were used to classify a sample as positive [75,93] (Appendix A). Thus, this technology does not seem to significantly improve the serological diagnosis of bTB compared to other available technologies.

On the other hand, SeraLyte-Mbv, a chemiluminescent test platform based on the single antigen MBP83, produced 89% sensitivity and 98% specificity when tested in cattle infected with *M. bovis*, MAP or *M. kansasii* [74] (Figure 3, Appendix A). However, there is only one study available in the literature that shows the promising results of this platform.

The DPP is a dual-path platform based on an immunochromatographic, point-of-care test format. Researchers have developed various DPP tests for bTB and assessed some of them in elephants suspected of having tuberculosis [135]. The first DPP test for bTB in cattle used the immunogenic proteins CFP10/ESAT6 and MPB70/MPB83, and showed a sensitivity of 62.2% and specificity ranging from 96.0% to 98.9% [95]. This test has subsequently yielded higher sensitivity (90.5%) by using a higher number of antigens (MPB70, MPB83, Rv2650c, Rv1463, and Rv3834c) [73] (Appendix A).

Another version of this platform includes the detection of IgM against the fusion protein MPB70-MPB83 and IgG against the fusion protein MPB70-MPB83-CFP10-Rv2650. The sensitivity and specificity of this test ranged between 71–100% and 95–100%, respectively [91]. Furthermore, these studies showed no reactivity in animals inoculated with nontuberculous mycobacteria and extremely low reactivity in MAP-vaccinated cattle (Appendix A) [91,101].

Regarding the reactivity to CFP10/ESAT6 and MPB70/MPB83, the DPP test showed a sensitivity of approximately 65% in TST-nonreactive cattle infected with *M. bovis* in a reduced number of samples (Appendix A) [95]. This finding encourages the use of this test as an ancillary method in regions with a high prevalence of bTB.

Another immunochromatographic test to detect antibodies against *M. bovis* is the LIONEX^®^ Animal TB Rapid Test, which includes the proteins MPB83, MPB70, ESAT6, and CFP10 (Figure 3, Appendix A). The sensitivity and specificity for this test were determined in only one assay using blood samples from TST/IGRA-positive and negative cattle and yielded values of 54% and 98.8%, respectively [92].

Research of an in-house lateral flow test evaluated in *M. bovis*-infected cattle, as confirmed by bacteriology, and in noninfected animals showed high sensitivity (83%) and specificity (99.4%) [100] (Appendix A). In herds with a high prevalence of bTB and bTB-free herds, a similar test showed high specificity but extremely low sensitivity, possibly due to the unknown bTB status of each sampled animal. However, this test efficiently detected *M. bovis*-infected cattle that had escaped TST diagnosis [94].

MAPIA, or Multi-antigen Print Immunoassay, is commonly used for antigenic protein screening [73,95,101,136] but less frequently used as a bTB diagnostic test [76] (Figure 3). This immunoassay involves printing multiple antigens onto a membrane. The analysis of bTB positivity by MAPIA using the criterion of the reaction of three antigens in PCR-confirmed *M. bovis*-infected and healthy cattle yielded a sensitivity and specificity of 69.39% and 90.27%, respectively, [76] (Appendix A).

Hemagglutination, the first method for detecting circulating antibodies against *M. bovis* in cattle with bTB, has largely fallen into disuse, despite efforts to develop improved versions of this method [137]. A latex bead agglutination assay has shown a sensitivity and specificity greater than 86% and 92%, respectively, by using MPB70 or ESAT6 as antigens [100] (Figure 3, Appendix A).

Finally, a fluorescence polarisation assay (FPA) using recombinant protein ESAT6 showed high sensitivity (>90%) but low specificity (<65%) in TST-positive/negative cows and buffaloes (Figure 3, Appendix A) [138]. When FPA was used with MPB70 as an antigen, the sensitivity and specificity were higher than for the ESAT6 test (Appendix A) [77]. Given the lack of robust gold standards for defining the bTB-positive and healthy groups [138], the low specificity observed for the ESAT6 antigen may be due to the inclusion of false negative samples in the bTB-negative group.

### 7.6. Antibody Detection in Milk and Urine Samples

The TST offers several benefits, including low-cost reagents and a straightforward procedure. However, it is labour-intensive because of the requirements of tuberculin injections in each animal and two visits to each farm: one for the injection and another to assess the reaction. By contrast, screening tests using milk samples are less labour-intensive and more cost-effective than testing individual animals with the TST.

Milk samples have been employed to control infectious diseases [139,140]. The milk ring test is a standard and official screening method for brucellosis [141]. Additionally, milk sample-based screening tests have been used for paratuberculosis (paraTB), a mycobacterial disease produced by MAP [142]. An ELISA using milk samples for paraTB was licensed in Germany and commercialised in the United States [143]. Studies on paraTB have shown a significant correlation between serum and milk ELISA results. Furthermore, the milk ELISA showed high sensitivity and specificity when compared to the serum ELISA as a reference [142].

In a pioneering study, Bo-Young Jeon et al. [144] explored the diagnosis of bTB using milk samples. They applied for the first time an ELISA based on the MPB70 antigen in milk, a test that achieved high sensitivity (87.8%) and specificity (97.7%) using TST as the reference standard. Notably, the reactivity of the milk samples showed a significant correlation with that of serum samples (Appendix A).

In 2011, Waters et al. [85] developed the now commercially available IDEXX ELISA and evaluated its performance employing milk samples. This research, together with the one from Buddle et al. [145], demonstrated that IDEXX detected antibodies in samples from infected cattle with a sensitivity of 50% (Appendix A). Furthermore, the ELISA’s sensitivity for milk and serum samples was the same. However, no significant differences were detected in the ELISA responses of bulk tank milk samples collected from infected and noninfected herds. The screening of bulk tank milk samples is unlikely to be useful in countries with low prevalence of *M. bovis* in cattle and large herd sizes [145].

Subsequent studies have assessed the performance of milk samples with different antigens. For example, Zhu et al. [146] evaluated milk samples with an ELISA coated with a chimaera protein formed by the IDEXX (MPB70/MPB83) and CFP10/ESAT6. In this case, the sensitivity with milk samples was similar to that obtained with the IDEXX antigens alone (Appendix A) [145]. Thus, the addition of ESAT6 and CFP10 does not seem to enhance the sensitivity of the ELISA test. On the other hand, similarly to the studies previously mentioned, serum-sample-based and milk-sample-based ELISA showed good agreement. Therefore, as milk samples are easier to collect, they may serve as a replacement for serum samples.

The previously mentioned Enferplex assay is another commercially available serology test used to diagnose bTB. This assay has demonstrated high sensitivity and specificity when using individual bovine milk samples, especially after boosting with PPDB (Appendix A) [147].

Apart from milk, urine is another sample easier to collect than serum. With this in mind, Zewude et al. [92] studied the performance of ALERE^®^ Determine TB LAM Ag, an immunoassay designed to detect lipoarabinomannan in urine, as a means to diagnose bTB. This kit, initially intended for diagnosing hTB disease in individuals with HIV/AIDS, can be used to detect LAM in the bovine urine. In active tuberculosis, LAM is released from both metabolically active and degrading bacteria. This glycolipid is then cleared through the kidneys and can be detected in urine [148]. The TB LAM Ag showed good diagnostic performance (Appendix A) and could be used as ancillary to TST or IFN-γ tests for bTB diagnosis [92]. However, subsequent research has found poor agreement between the urine LAM test and other routine bTB tests (Appendix A) (IGRA, necropsy, histology, culture, PCR) [149].

Altogether, milk samples generally perform well and can be an alternative to serum for diagnosing bTB. However, the use of urine for bTB diagnosis is controversial and requires further study.

### 7.7. Serological Diagnosis in Other Animals

*M. bovis* can affect several wildlife species, which can act as reservoirs of the disease. This characteristic presents significant economic and animal health problems and poses a greater challenge in combating the dissemination of the disease. Despite the difficulties in obtaining samples or capturing the infected species, the diagnosis of bTB in wildlife is a major challenge. As mentioned before, the TST has shown adequate sensitivity and specificity when applied for bTB diagnostic [150,151]. In this section, we summarise the advances in serological diagnosis of wild animals implicated in the persistence of *M. bovis* infection in herds.

Researchers have reported wildlife bTB reservoirs in different regions [151]. For example, in the United Kingdom, badgers and wild deer species spread and maintain bTB. To tackle this problem, researchers have developed a test, known as the CervidTB STAT-PAK, based on the antigens MPB83, ESAT6 and CFP10 for detecting *M. bovis* and *M. tuberculosis* antibodies [152]. The sensitivity of this test ranges from 54.5% to 85.7% and this variation seems to depend on the deer species evaluated [152,153,154].

A study compared two lateral flow tests, CervidTB STAT-PAK and DPP VetTB (see above), in farmed red deer experimentally or naturally infected with *M. bovis* [155,156]. Both tests exhibited equally high sensitivity, although the DPP VetTB test had slightly higher specificity (91.4%) compared to the CervidTB STAT-PAK test (83.8%).

An ELISA based on PPDB for detecting bTB in European wild boars that was first assessed in captive wild boars sensitised with inactivated bacterial antigens showed a sensitivity of 72.60% and a specificity of 96.43% for the best cut-off [157].

Four ELISA tests and a lateral flow immunochromatographic test were simultaneously evaluated in *M. bovis*-naturally infected domestic free-range pigs with confirmed infection (according to qPCR or *M. bovis* isolation) [158]. Among the methods evaluated, the ELISA combining MPB83 and MPB70 and the coloured-latex-based immunochromatographic lateral flow dipstick (LFD) using MPB83 as the sole antigen showed the best sensitivity (78% and 74.6%, respectively) and specificity (100% and 98.9%, respectively).

More recently, a research group evaluated the multi-antigen assay TB Luminex multiplex test in pig sera. This test includes the antigens MPB83, MPB70, CFP10 and ESAT6. The inclusion criterion for positive samples was the evidence of grossly visible lesions. The study found that MPB83 was immunodominant, while ESAT6 showed the lowest performance [159]. Similarly, the commercial test DPP VetTB Assay, tested in the three suid species, showed higher reactivity to MPB83 than to CFP10/ESAT6. However, the combination of responses to MPB83 or CFP10/ESAT-6 yielded the highest sensitivity (80.4–95.2%) and specificity (91.4–100%) [160].

Another serologic test used in South American Camelids such as alpacas and llamas, which are highly susceptible to bTB by *M. bovis* and *M. microti*, consists of an ELISA based on P22, a multiprotein complex obtained by affinity chromatography from the PPDB [161]. In these animals, the test showed high efficiency and sensitivity.

In conclusion, antibody detection techniques for disease diagnosis have the advantage of requiring low-complexity laboratories for their implementation, being more economical, allowing antemortem evaluation, and generally not requiring the treatment of animals for their implementation. ELISA-based humoral diagnosis of bTB has proven to be robust, simple, and economically affordable. Newer diagnostic methods are simpler to perform than ELISA and can be used directly at the animal’s location. Among these recent technologies, DPP has shown remarkable performance. In addition, serological tests are more convenient than cell-mediated-based tests for detecting bTB in wildlife species, since the samples are easier to collect, and the tests are applicable for varied species.

### 7.8. Factors Influencing the Humoral Response to M. bovis in Cattle

Serological studies conducted in parallel across different geographical regions or countries have shown variable sensitivity and specificity values in the immune response to bTB [72,95]. These findings suggest that this humoral response is modulated by factors such as the prevalence of the disease [162], the stage of disease progression [163], the presence of environmental mycobacteria or other pathogenic mycobacteria [113], and genetic variability among individuals [164,165,166]. Another factor that influences the specific production of antibodies against bTB is the immune status of the animals. As previously mentioned, vaccination or the TST generates a booster effect on the humoral response [99]. This particular aspect of the immune response is commonly used to improve the sensitivity of tests that measure the production of antibodies against *M. bovis*. However, this booster effect has the limitation of confining the response solely to the antigens present in PPDB or experimental vaccines. In the case of vaccination with BCG, for example, a booster does not enhance the production of antibodies against relevant antigens such as ESAT6 and CFP10. On the other hand, the AN5 strain used for the production of PPDB has shown lower levels of transcription of numerous genes, including *mpb83* and *mpb70* compared to field strains [167].

Therefore, the combination of these factors, which includes the genetics of the animals, the course of tuberculosis, the treatments applied (TST or experimental vaccines), and environmental components, may explain the different performances achieved by the same test in various trials around the world (Appendix A).

## 8. Conclusions

For many years, research on the immunology of human and animal tuberculosis has focused on studying the cellular response. However, researchers have recently started unravelling overlooked aspects of the humoral and B cell immune responses in these diseases. Novel high-throughput technologies have allowed the discovery of various roles of B cells and antibodies in immunity against hTB. Despite the scarce knowledge on the roles of these immune responses in bTB, the new insights and technologies applied to studying the B cell-mediated response in hTB would be transferable to animal tuberculosis research, given the similarity between the two diseases.

As previously mentioned, antibody production and possibly B cell functions are reduced during the early stages of *M. bovis* infection. In addition, during active hTB, certain subpopulations of B lymphocytes are altered compared to individuals with latent tuberculosis [27,168]. These findings and others lead us to question whether pathogenic mycobacteria deploy mechanisms to suppress certain B cell response profiles and thereby establish a successful infection. Understanding this response and deciphering these mechanisms will undoubtedly contribute to the development of better vaccines and treatments.

Considerable progress has been made in studying the antibody-mediated response for diagnostic purposes in livestock. As animals reach advanced stages of the disease, the humoral response increases and the cellular response decreases. By exploiting this particularity, researchers have developed numerous diagnostic tests for tuberculosis with extremely high sensitivity and specificity and with the ability to identify infected animals that escape traditional diagnostic tests. In addition, the replacement of whole antigens, such as culture supernatant proteins, PPDB, or *M. bovis* cell extract, with recombinant antigenic proteins has improved the sensitivity of ELISA tests and other diagnostic platforms. Furthermore, new immunochromatographic technologies based on multiple recombinant antigens and diverse types of immunoglobulins have facilitated the development of point-of-care tests to reinforce epidemiological surveillance in livestock production regions with limited access to veterinary services.

## Figures and Tables

**Figure 1 antibodies-13-00084-f001:**
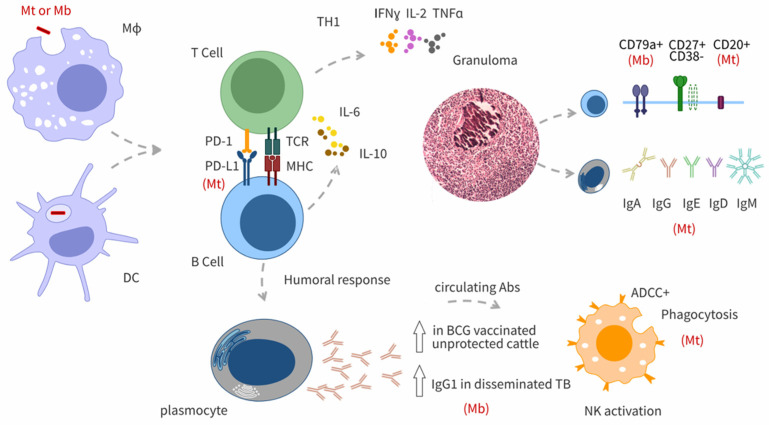
Relevant aspects of the immune response in tuberculosis. Macrophages (Mo) and dendritic cells (DC) are infected by *M. tuberculosis* (Mt) or *M. bovis* (Mb) and subsequently present antigens to T and B cells priming the adaptive immune response. T cells then adopt a Th1 phenotype, characterised by the production of IFN-ɣ, IL-2, and TNF-α. B cells serve as antigen-presenting cells and interact with T cells via specific receptor-ligand molecules and cytokine signalling. Specific B cell subpopulations (CD79a+, CD27+CD38-, CD20+) identified in lung tissue from BCG-vaccinated cattle (Mb), and non-human primates and patients infected with *M. tuberculosis* (Mt) are associated with the granuloma. B cells differentiate into plasmocytes, which release antibodies targeting Mt. Specific IgA, IgG, IgE, IgD, and IgM antibodies were detected in lung tissues of patients with tuberculosis. The figure highlights the crucial roles of antibodies in tuberculosis, including antibody-dependent cellular cytotoxicity (ADCC+), antibody-dependent activation of natural killer (NK) cells, and antibody-dependent cellular phagocytosis.

**Figure 3 antibodies-13-00084-f003:**
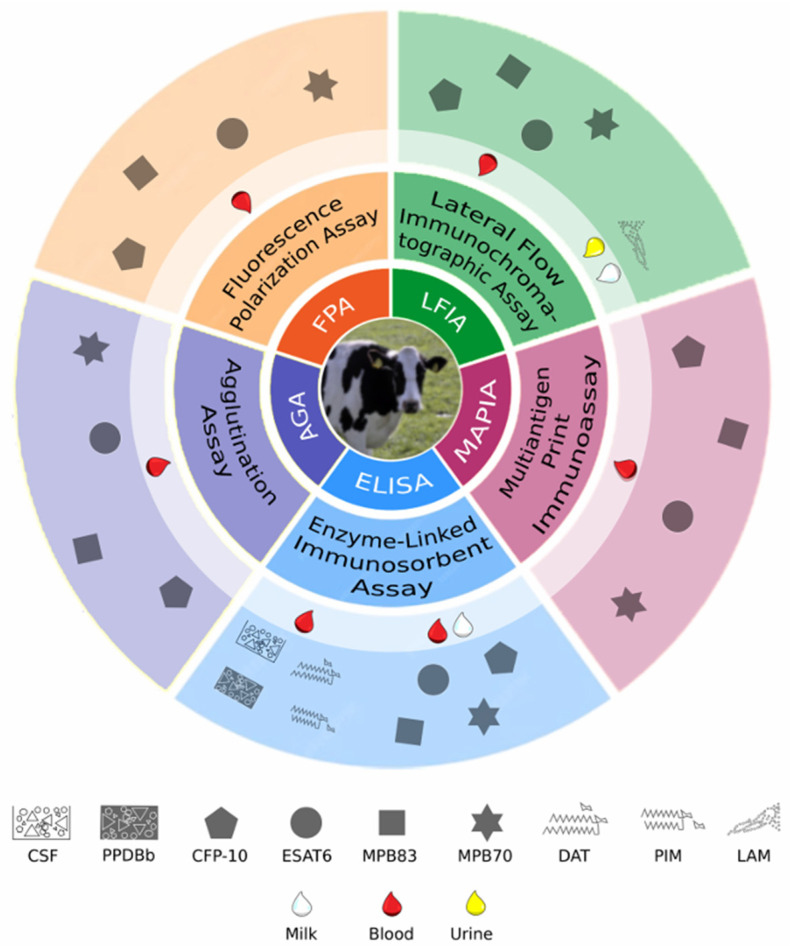
Key platforms for antibody detection in tuberculous animals. The specific antigens used in each test are schematically illustrated. Red, white, and yellow droplets represent the types of samples tested in each assay—blood, milk, and urine, respectively.

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
