# Peer review of "B Cell and Antibody Responses in Bovine Tuberculosis"

_2073-4468, 2024, doi:10.3390/antib13040084_

Round 1

Reviewer 1 Report

Comments and Suggestions for Authors

I found this review paper very insightful, and I appreciate that you have tried to cover and compare a wide range of studies and data, but this sometimes made the paper hard to follow. The use of tables and figures did not help with the flow of the paper and should be readdressed.

Please see attached file for breakdown of comments.

Comments on the Quality of English Language

Overall the use of English language was very good. General comments were around the overuse of abbreviations which sometimes made the reading flow difficult, and inconsistencies when using abbreviations. these have all been documented in the attached file. The use of figures and tables also interrupted the flow of the text and should be reconsidered.

Please see attached file for breakdown of comments.

Author Response

Overview

This review intends to provide a comprehensive overview of the use of antibodies against bTB for diagnostic purposes and the role of antibodies and B cells in the immunopathology and host defence against bTB. They do this by discussing advances the study of antibodies in human TB and compare against the contradictory evidence available on the humoral response associated with bovine TB.

My main comment would be that it leans more towards a review of the use of antibodies in the diagnostic setting, in particular the comparative use of antibody detection in serum, milk, and urine. Whilst this is very useful it would have been nice if there had been more discussion/hypothesis around the direct role of B cells and antibodies. Perhaps the authors could summarise with a diagram highlighting how the elements of the humoral and cell- mediated immune response are interacting and the potential outcomes on disease severity.

Thank this reviewer very much for your comment. We have added a new figure (Fig. 1) that aims to illustrate the primary immune mechanisms involved in the interaction between mycobacteria and the host.

My general comments/observations on the paper are that there are far too many abbreviations, and inconsistencies in use of abbreviations, and that the use of figures and tables make it hard to follow the paper at times.

We agree with these concerns and we moved table 2 to the supplementary file. Regarding the previous Figure 2, we realized that the figure was incorrectly submitted and lacked important information. We have now provided a corrected version, which is included as Figure 3. Additionally, we have removed Table 1..  

Other than that, I think it is a very useful review paper that provides substantial evidence for requirement for further research in this field to better understand the importance of the humoral immune response in bovine tuberculosis.

Thanks!

Edits:

  1. Watch abbreviations

Abbreviations were revised.

  1. Line 59 - antigen Sp. ‘Sp’ not referenced as an abbreviation until line 236.

This was corrected. 

  1. Active tuberculosis ‘ATB’ all caps but in line 68 written as ‘ATb’.

According to the suggestion mentioned above, these abbreviations were removed. 

  1. Perhaps consider using the words latent and active rather than the abbreviations LTI

and ATB to make smoother reading.

We agree with this suggestion and change the abbreviations for the whole words.

  1. Line 67 - THP1 monocytes could do with an explanation that these are a human cell

line.

This clarification was added.

  1. Line 77 to Line 97 – the use of B cells vs B-cells inconsistent

This inconsistency was corrected. 

  1. Figure 1. integration into the text – would this be better as diagram? Feels disjointed

from the text without due reference or explanation.

  1. Figure 1 - Kinetics of the humoral response title on top right-hand box, implies a

timeline study of responses so could be illustrated with time points of Ab responses or change the title to something along the lines of outcomes of different CMI/Ab response pathways.

  1. Figure 1 – perhaps reference the paper for each box and reference the figure in the relevant text section?

The three suggestions regarding figure 1 were taken into account and the figure (now figure 2) was modified and integrated into the text.

  1. Line 116 - too large a space between techniques and ref (30).

The space was eliminated. 

  1. Table 1 vs. Table S1 - Why is there a Table 1 and then a supplementary table 1

showing almost the same data? Do they need both? Table S1 is compiled in much better format than Table 1 and I would use it to replace Table 1.

Table 1 was removed from the text and replaced by Table S1.

  1. Note Table 1 uses 5 x mis-spelt uses of Multi-antigens from Multipleantigens to Mulltipleantigens.

These errors were corrected from table S1.

  1. Table 1 - is a lot of data to take in with constant scanning to the end of the table to check each abbreviation. Can this be changed?

Table 1 was removed and replaced by table S1 which does not have so many abbreviations. 

  1. Line 284 - MPB83: 75-100%; %Table 1, Table S1 remove % symbol before the word Table.

The symbol was removed.

  1. Figure 2 – hard to follow, requires more explanation. What is the point of it? What are they trying to show? Is the list of antigens with symbols supposed to be displayed on the figure? It doesn’t make sense.

Figure 2 was replaced by an improved (and correct) version. 

  1. Table 2 not referenced in the text – should it go in section 3.6?

Table 2 was sent to supplementary material (table S2) and now Table S2 is referenced in the text.

  1. Line 298-299 check font for the numbers in parentheses, inconsistent.

This was corrected.

  1. Line 311 – insert % after Se and Sp values.

The symbol was inserted.

Suggestions

I found this review paper to be a very informative compilation of studies, which is difficult since the authors were reviewing and comparing a wide range of data covering many different animal studies, experimental platforms, and uncontrolled variables.

Taking this into account I think the paper would benefit by the authors commenting on differences in humoral responses potentially being related to the varied types of studies. Such as: location of study (TB responses vary in different countries), host species (some hosts have higher susceptibility), impact of environment exposure, Study design (experimental vs. natural infections studies), impact of using different BCG vaccine strains, differences in tuberculin skin testing. Also, many of the studies looked at the humoral response following a skin test or BCG vaccination and therefore, subsequent antibody detection methods may not recognise key proteins found in the M. bovis genome but absent from PPDB or BCG.

Perhaps they could also comment on different mycobacterium species in bovine TB, such as the recognition of M. orygis in Mycobacterium Tuberculosis complex pathogen group (MTBC) which is found in Sout East Asia, India and Africa regions and may be responsible for more dairy cattle infections than M. bovis.

Thank you very much again for your comments. We have tried to address the reviewer's suggestions by adding the following text “In addition, other members of the Mycobacterium tuberculosis complex, such as Mycobacterium orygis, can also cause bTB (2,3)” to the introduction and including a new section titled “Factors influencing the humoral response to M. bovis in cattle”.

Reviewer 2 Report

Comments and Suggestions for Authors

 Review 1

B cell and antibody responses in bovine tuberculosis 2

Laura Inés Klepp 1,2*, Federico Carlos Blanco 1,2*, María Mercedes Bigi 3, Cristina Lourdes Vázquez 1,2, Elizabeth 3 Andrea García 1,2, Julia Sabio y García 1,2* and Fabiana Bigi 1,2*Ω

The authors describe the role of antibodies in bovine tuberculosis. They rightfully indicate that this is an understudied area of tuberculosis in human and even more so in bovine tuberculosis research and control.

Introduction

The introduction is a little unbalanced / not optimally organized in my opinion. A clear differentiation should be made between immune responses which are generated through natural infection and immune responses which are generated through interventions such as vaccination or in vivo diagnostic procedures such as skin testing. It is also important to note that immune response are generally evaluated using blood samples for both antibodies and CMI. There may (is likely) a discrepancy between what is measured here and local tissue responses. Authors realize this (eg line 168-182) but it is currently buried in other sections.

Generalized statements such as line 41 (While the cell-mediated immune response is crucial in the defence against bTB,) should be avoided because they are not per se true. The reason for this is that in indirect diagnostic procedures in bovine TB we (as indicated) rely on measuring cell mediated immunity (CMI) parameters (skin test, IGRA) and we use the outcome to identify infected animals and cull them. In this way we use the CMI as a measure of disease severity. We do not use it as a measure of protective immunity otherwise we would be deliberately removing animals with protective immune responses from the population.

Similar discussions can be found in literature with respect to human tuberculosis and CMI.

In the same line of reasoning we should also not use generalized statements regarding antibody responses as being associated with disease without the proper nuances. In terms of indirect diagnostic tests all immune measures are used as indicators of disease and used to either control or treatment actions in bovine and human TB respectively.

We know by now that pathogenic mycobacteria are masters in manipulating host immune responses and this should be indicated. As a result we should be very careful in trying to interpret responses observed during disease in terms of protective of permissive as it is a result of the complex interaction between the host and the pathogen. This should be reflected in the introduction and discussion.

 CMI and antibody responses initiated through vaccination are inherently different because they are generated without the immunomodulatory actions of the pathogenic mycobacteria. Therefor in terms of timing, (tissue)location, magnitude and quality of the response the outcome maybe different despite superficial similarities with responses measured in circulation during disease. In addition (with the potential exception of experimental murine models) there is a large variation in the host population based on genetic, epigenetic and environmental (including the mycobacteria themselves) factors determining the outcome of infection. Typically a small fraction of a population appears naturally resistant, most infections lead to latent infections and only a small fraction develops clinical – symptomatic tuberculosis. In human populations there is a major role for immunosuppressive co-infections (eg HIV) or malnutrition which is not so prominent in bovine tuberculosis.

 These nuance and details are very important and not outlining this properly is one of the reasons that the literature is highly contradictive regarding interpretation of adaptive immune responses during mycobacterial infections and vaccination trials.

 I strongly urge the authors to revise the introduction and discussion accordingly.

Chapter 2

See remarks for introduction: separate infection and vaccination studies more clearly

 Line 100-104: please provide (a) reference(s) for this paragraph

Chapter 3

Line 227: …overly sensitive… is unclear, do the authors mean it is too sensitive or very sensitive?

 Line 270-277: it is actually quite remarkable that there is an almost absolute skewing of the B cell response towards MPB70/MPB83 proteins considering there are some 4000 protein ORFs in the genome many of which are expressed when considering e.g. CFP preparations. In other mycobacterial infections and many other bacterial infections a very broad range of bacterial proteins is immunogenic and a diverse response can be observed.

Line 267-277: please provide a reference for this statement. Has it been documented that MPB70/MPB83 concentrations are exceptionally high compared to other proteins? Can you provide the evidence for this? Also in light of the statements at line 314-327?

Chapter 4

 Given the lack of knowledge on the immunobiology of B cells and antibodies in tuberculosis should we consider the possibility that M. bovis / M. tuberculosis actively manipulate the host B cell response upon infection?

Author Response

Introduction

The introduction is a little unbalanced / not optimally organized in my opinion. A clear differentiation should be made between immune responses which are generated through natural infection and immune responses which are generated through interventions such as vaccination or in vivo diagnostic procedures such as skin testing. It is also important to note that immune response are generally evaluated using blood samples for both antibodies and CMI. There may (is likely) a discrepancy between what is measured here and local tissue responses. Authors realize this (eg line 168-182) but it is currently buried in other sections.

Generalized statements such as line 41 (While the cell-mediated immune response is crucial in the defence against bTB,) should be avoided because they are not per se true. The reason for this is that in indirect diagnostic procedures in bovine TB we (as indicated) rely on measuring cell mediated immunity (CMI) parameters (skin test, IGRA) and we use the outcome to identify infected animals and cull them. In this way we use the CMI as a measure of disease severity. We do not use it as a measure of protective immunity otherwise we would be deliberately removing animals with protective immune responses from the population.

Similar discussions can be found in literature with respect to human tuberculosis and CMI.

In the same line of reasoning we should also not use generalized statements regarding antibody responses as being associated with disease without the proper nuances. In terms of indirect diagnostic tests all immune measures are used as indicators of disease and used to either control or treatment actions in bovine and human TB respectively.

We know by now that pathogenic mycobacteria are masters in manipulating host immune responses and this should be indicated. As a result we should be very careful in trying to interpret responses observed during disease in terms of protective of permissive as it is a result of the complex interaction between the host and the pathogen. This should be reflected in the introduction and discussion.

 CMI and antibody responses initiated through vaccination are inherently different because they are generated without the immunomodulatory actions of the pathogenic mycobacteria. Therefor in terms of timing, (tissue)location, magnitude and quality of the response the outcome maybe different despite superficial similarities with responses measured in circulation during disease. In addition (with the potential exception of experimental murine models) there is a large variation in the host population based on genetic, epigenetic and environmental (including the mycobacteria themselves) factors determining the outcome of infection. Typically a small fraction of a population appears naturally resistant, most infections lead to latent infections and only a small fraction develops clinical – symptomatic tuberculosis. In human populations there is a major role for immunosuppressive co-infections (eg HIV) or malnutrition which is not so prominent in bovine tuberculosis.

These nuance and details are very important and not outlining this properly is one of the reasons that the literature is highly contradictive regarding interpretation of adaptive immune responses during mycobacterial infections and vaccination trials.

 I strongly urge the authors to revise the introduction and discussion accordingly.

We thank the reviewer for all his/her comments regarding our manuscript. We have addressed all these concerns and have restructured the manuscript accordingly. Part of the introduction discussing the background on B cell responses in human tuberculosis has been moved to a separate section. Additionally, we have included an overview of the immune responses in bovine tuberculosis in the introduction, distinguishing between those induced by natural infection and those induced by vaccination. We have also added a section describing ante-mortem diagnostic methods to address concerns about the differences between the types of samples used in diagnostic methods.

We also agree with the reviewer´s viewpoint regarding the dogma that places central importance on cellular immune responses for controlling tuberculosis. For this reason, we have removed from the text any statement asserting a single position on this highly debated aspect among researchers.

Chapter 2

See remarks for introduction: separate infection and vaccination studies more clearly

They have been separated.

Line 100-104: please provide (a) reference(s) for this paragraph

 Thanks for pointing out the omission. A reference has been included.

Chapter 3

Line 227: …overly sensitive… is unclear, do the authors mean it is too sensitive or very sensitive?

We apologize for the ambiguous word. It has been corrected.

 Line 270-277: it is actually quite remarkable that there is an almost absolute skewing of the B cell response towards MPB70/MPB83 proteins considering there are some 4000 protein ORFs in the genome many of which are expressed when considering e.g. CFP preparations. In other mycobacterial infections and many other bacterial infections a very broad range of bacterial proteins is immunogenic and a diverse response can be observed.

We believe we have already addressed this issue in pages 13-14 by highlighting the importance of also considering cellular proteins of M. bovis as diagnostic antigens.

Line 267-277: please provide a reference for this statement. Has it been documented that MPB70/MPB83 concentrations are exceptionally high compared to other proteins? Can you provide the evidence for this? Also in light of the statements at line 314-327?

Thank you for the comment. While it has been demonstrated that M. bovis expresses high levels of the MPB70 protein, only the MPB83 protein is among the 10 most abundant proteins in the PPDB, alongside ESAT6 and CFP10. However, a notable feature of MPB70 and MPB83 is their high resistance to heat treatment. The new version of the manuscript includes this information and all the corresponding references.

Chapter 4

Given the lack of knowledge on the immunobiology of B cells and antibodies in tuberculosis should we consider the possibility that M. bovis / M. tuberculosis actively manipulate the host B cell response upon infection?

Thanks for the suggestion. We found this rationale very compelling. However, we were unable to find specific data on the evasion of B cell responses in tuberculosis. We have included a text in the conclusion section with an open question addressing this issue.

Round 2

Reviewer 2 Report

Comments and Suggestions for Authors

Thank you for carefully considering my comments and revising the manuscript.